# New Compounds with Bioisosteric Replacement of Classic Choline Kinase Inhibitors Show Potent Antiplasmodial Activity

**DOI:** 10.3390/pharmaceutics13111842

**Published:** 2021-11-02

**Authors:** Francisco José Aguilar-Troyano, Archimede Torretta, Gianluca Rubbini, Alberto Fasiolo, Pilar María Luque-Navarro, María Paz Carrasco-Jimenez, Guiomar Pérez-Moreno, Cristina Bosch-Navarrete, Dolores González-Pacanowska, Emilio Parisini, Luisa Carlota Lopez-Cara

**Affiliations:** 1Department of Pharmaceutical and Organic Chemistry, Faculty of Pharmacy, Campus of Cartuja, 18071 Granada, Spain; francisco.aguilar_troyano@uni-wuppertal.de (F.J.A.-T.); gianluca.rubbini@unimi.it (G.R.); albertofasiolo@gmail.com (A.F.); pilarluque@ugr.es (P.M.L.-N.); 2Center for Nano Science and Technology @PoliMi, Istituto Italiano di Tecnologia, Via Pascoli 70/3, 20133 Milano, Italy; archimede.torretta@iit.it; 3Department of Biochemistry and Molecular Biology I, Faculty of Sciences, 18071 Granada, Spain; mpazcj@ugr.es; 4Department of Biochemistry and Molecular Pharmacology, Institute of Parasitology and Biomedicine “López-Neyra”, Spanish National Research Council, Parque Tecnológico de Ciencias de la Salud, Avenida del Conocimiento 17, 18016, Granada, Spain; guiomar@ipb11.ipb.csic.es (G.P.-M.); cristinabosch@ipb.csic.es (C.B.-N.); 5Department of Organic Synthesis, Latvian Institute of Organic Synthesis, Aizkraukles 21, LV-1006 Riga, Latvia

**Keywords:** antimalarial drug, choline kinase inhibition

## Abstract

In the fight against Malaria, new strategies need to be developed to avoid resistance of the parasite to pharmaceutics and other prevention barriers. Recently, a Host Directed Therapy approach based on the suppression of the starting materials uptake from the host by the parasite has provided excellent results. In this article, we propose the synthesis of bioisosteric compounds that are capable of inhibiting *Plasmodium falciparum* Choline Kinase and therefore to reduce choline uptake, which is essential for the development of the parasite. Of the 41 bioisosteric compounds reported herein, none showed any influence of the linker on the antimalarial and enzyme inhibitory activity, whereas an effect of the type of cationic heads used could be observed. SARs determined that the thienopyrimidine substituted in 4 by a pyrrolidine is the best scaffold, independently of the chosen linker. The decrease in lipophilicity seems to improve the antimalarial activity but to cause an opposite effect on the inhibition of the enzyme. While potent compounds with similar good inhibitory values have been related to the proposed mechanism of action, some of them still show discrepancies and further studies are needed to determine their specific molecular target.

## 1. Introduction

Malaria is a disease caused by *Plasmodium*, a parasite that involves cyclical infection between the female mosquito Anopheles and human erythrocytes. It is in the erythrocyte, where parasite division and sexual differentiation take place, thus allowing the diffusion of the pathogen in the bloodstream of the host and causing anaemia and organ failure in severe cases.

The world regions at risk of the disease are sub-Saharan Africa, South-East Asia, Eastern Mediterranean, Western Pacific, and the Americas, a situation that affects nearly half of the world population and that continues to worsen with climate change.

In 2019, there were 229 million cases of malaria, the most fragile population being children under the age of five [1].

Owing to the problem of resistance development to antimalarials, insecticides and other prevention barriers, the scientific community has started to look for new strategies to block parasite progression and infection.

Vial et al. first proposed a strategy based on the suppression of the availability of the host’s nutrients to the parasite [2]. They exploited the fact that once the parasite has infected the erythrocyte, there is a cycle of asexual division in which each member of the new progeny needs to be packed in a lipidic membrane. As the erythrocyte is a cell with no nucleus, it is the enzymatic machinery of the parasite that converts the host starting materials into the necessary phospholipids.

For this reason, the uptake inhibition of phospholipid precursors such as choline or ethanolamine can become a successful host-directed therapy (HDT) [2]. Other studies have focused on the inhibition of transporters necessary for the salvage of purine bases [3] and glucose [4], as the parasite is unable to carry out their *de novo* production.

Among the strategies based on differences between the host and the parasite metabolism, recent studies have shown that the reduction of the accessibility to phospholipids makes the parasite reproduce in a sexual rather than in an asexual way [5]. This reduces malaria mortality because of the lower erythrocytic infection by merozoites. Sexual differentiation results in the formation of gametocytes that remain in the bloodstream and disappear from the circulation if not taken up by a mosquito.

We have previously studied the enzymatic inhibition of *Plasmodium falciparum* Choline Kinase (*Pf*CK), a monomeric protein whose crystallographic structure has been recently elucidated in complex with Mg^2+^ atoms and ATP [6]. In the structure, the catalytic pocket is very similar to its human CK equivalent, showing conserved hydrophobic and negatively charged residues where the positive charge of choline is stabilized.

Moreover, we have previously reported that biscationic inhibitors could show different mechanisms of action depending on their structure. For instance, biscationic (Figure 1) compounds with an aliphatic linker (**BR23**, **BR25**) can inhibit the ethanolamine kinase activity of *Pf*CK while other compounds **(BR31, BR33, 10a-l)** inhibit the choline kinase activity of the enzyme [7,8,9].

Some monocationic compounds have also been evaluated for their antitumor and antimalarial activity [8,10,11,12,13]. Recently, it was described that the different conformation adopted by the enzyme depending on whether the substrate is choline or ethanolamine could determine the different mode of binding adopted by the compounds in the CK pocket [7].

Two interesting reviews by Vial et al. [14,15] describe the acquisition of glycerophospholipids in *Plasmodium*, showing how the Kennedy pathway involves different mechanisms for the formation of phosphocholine (PC) and phosphoethanolamine (PE), the main membrane phospholipids, and how this pathway has autoregulatory mechanisms. In 2020, this review was updated by *Mulaw* [16] with new and interesting targets.

Clearly, exciting data validate *Pf*CK as a drug target in malaria disease. *Plasmodium* species parasites need to grow and multiply fueled by precursors supplied by the host. Among these precursors of particular importance are choline and ethanolamine used by the parasite to synthesize its phospholipids, mainly PC, the major phospholipid in *Plasmodium* membranes like in the rest of eukaryotes. It has been demonstrated that the specific inhibition of *Pf*CK disrupts its Kennedy pathway, resulting in parasite death. For this purpose, many chemical compounds targeting this enzyme have been developed.

Bioisosteric changes have also been shown to be very useful in reducing the toxicity or modifying the activity of many lead compounds, which also leads to pharmacokinetic alteration. Therefore, the bioisosterism strategy has been widely used to develop new therapeutic substances used as receptor antagonists or agonists, enzyme inhibitors and antimetabolites.

The substitution of quinoline by thienopyridine has been extensively studied in different pathologies related to kinase inhibition. Among them, a large number of thienopyrimidine derivatives have been published where their antitumor activity has been enhanced by inhibiting multiple enzymes as well as by modulating the activity of many receptors [3,17,18,19,20,21,22,23,24].

These observations, together with the low toxicity described for some of these previous compounds [3,17,18,19,20,21,22,23,24], have led us to evaluate all compounds described in this work as antimalarials and to study their possible mechanism of action as inhibitors of *Pf*CK as the first approach in our study.

In the search for improving antimalarial activity, we present here the synthesis and biological evaluation of 41 compounds in which bioisosteric changes have been introduced on the cationic heads. While retaining the linkers used so far between the two heads, three types of cationic heads were chosen: thieno[3,2-b] pyridine, thieno [2,3-d] pyrimidine and thieno[3,2-d] pyrimidine (Figure 2 and Table 1). Those structures were selected in order to study the effect of introducing bioisosteric changes over the antimalarial activity and the enzymatic inhibition. In addition, these new derivatives were developed with the aim to determine whether the replacement of quinoline and pyridine as cationic heads of the classical compounds would enable the molecules to adopt the correct geometry for activity. Furthermore, the presence of the additional electron pair in the new N and S atoms would lead to an increase in the affinity for the target through the formation of new hydrogen bonds, together with an increase in bioavailability resulting from its higher solubility.

## 2. Materials and Methods

### 2.1. Chemistry

Melting points were taken in open capillaries on a Stuart Scientific SMP3 electrothermal melting-point apparatus and were uncorrected. Analytical thin-layer chromatography (TLC) was performed using Merck Kieselgel 60 F254 aluminium plates and visualized by UV light or iodine. All evaporation occurred *in vacuo* with a Büchi rotary evaporator (Büchi, Barcelona, Spain) and the pressure was controlled by a Vacuubrand CVCII apparatus (Vacuubrand, Wertheim, Germany). For flash chromatography, Merck silica gel 60 with a particle size of 0.040–0.063 mm (230–400 mesh ASTM) was used (Merck KGaA, Darmstadt, Germany). For NMR spectra were used: 500 MHz ^1^H and 126 MHz ^13^C NMR Varian Direct Drive; a 400 MHz ^1^H and 101 MHz ^13^C NMR Varian Direct Drive spectrometers at room temperature (Varian Medical Systems, Palo Alto, CA, USA). Chemical shifts (δ) are quoted in parts per million (ppm) and are referenced to the residual solvent peak. Spin multiplicities are given as s (singlet), bs (broad singlet), d (doublet), t (triplet), ddd (doublet doublet doublet), dd (doublet doublet), dt (doublet triplet), q (quadruplet) and m (multiplet). High-resolution NanoAssisted Laser Desorption/Ionization (NALDI-TOF) or Electrospray Ionization (ESI-TOF) mass spectra were carried out on a Bruker Autoflex or a Waters LCT Premier Mass (Agilent Scientific Instruments, Santa Clara, CA, USA). Organic solutions were dried over anhydrous Na_2_SO_4_. Solvents and reagents that are commercially available were purchased from Aldrich (Sigma-Aldrich, Burlington, MA, USA) or Alfa Aesar (Johnson Matthey Company London, England, UK) and were used without further purification unless otherwise noted.

### 2.2. Cloning, Protein Expression and Purification of PfCK

A truncated form of PfCK (Δ79-440) featuring an *N*-terminal 6× His-tag was purchased from Genescript (Piscataway, NJ, USA), cloned into a pET-28a vector and transformed into *Escherichia coli* BL21 (DE3) Star cells (Invitrogen, Carlsbad, CA, USA). The transformed bacteria were cultured in Luria-Bertani (LB) medium at 37 °C until OD_600_ = 0.6 and the expression was induced with 1 mM isopropyl β-D-1-thiogalactopyranoside (IPTG). The cell culture was grown overnight at 20 °C and 180 rpm. Cells were pelleted by centrifugation at 10,000 rpm, resuspended in 50 mM Tris/HCl pH 7.5, 500 mM NaCl, 0.2 mM phenylmethylsulphonyl fluoride (PMSF, Sigma Aldrich, St. Luis, MO, USA), DNase (Sigma Aldrich, St. Luis, MO, USA), 0.5 mM β-mercaptoethanol (Sigma Aldrich, St. Luis, MO, USA) and sonicated. Separation of the soluble from the insoluble fraction was obtained by centrifugation at 15,000 rpm and 4 °C. The target enzyme was purified by a two-step purification protocol. In the first step, the cell lysate was incubated for 45 min with Ni-NTA resin (Qiagen, Venlo, The Netherlands) in a gravity column. Then, the column was washed with 40 column volumes (CV) of 50 mM Tris/HCl pH 7.5, 300 mM NaCl, 10 mM imidazole and 1 CV of 50 mM Tris/HCl pH 7.5, 300 mM NaCl, 40 mM imidazole. Finally, the His-tagged enzyme was eluted from the column using a 50 mM Tris/HCl pH 7.5, 300 mM NaCl, 400 mM imidazole buffer. A subsequent size-exclusion chromatography step was implemented to further polish the sample using a HiPrep 26/60 Sephacryl 100 HR column (GE Healthcare, Little Chalfont, Buckinghamshire, UK) previously equilibrated with 20 mM Tris/HCl pH 7.5, 150 mM NaCl buffer. This second purification step afforded a pure protein sample. The protein production yield was estimated to be nearly 2.5 mg of recombinant enzyme per liter of bacterial culture [7].

### 2.3. Enzymatic Assay for the Calculation of the IC_50_s of the Selected Compounds

A spectrophotometric 96-well plate assay for *Pf*CK activity was established based on an assay previously described [7]. The reaction mix was composed as follows: 100 mM Tris/HCl pH 7.5, 100 mM KCl, 10 mM MgCl_2_, 0.5 mM phosphoenolpyruvate, 0.25 mM NADH, 4 U of pyruvate kinase, 5 U lactate dehydrogenase, 2 mM ATP, 4 mM choline chloride and different concentrations of the selected compounds. All the necessary compounds for the assay were purchased from Sigma Aldrich (St. Luis, MO, USA) and the final assay volume was 200 μL. The enzymatic reaction was initiated by the addition of the enzyme at less than 10 nM concentration and the reaction rates were measured by monitoring the decrease in the absorbance at 340 nm for 20 min at room temperature using a Spark10M instrument (Tecan). Finally, the IC_50_ values for each compound were determined by plotting the enzyme fractional activity (f.a.) against the logarithm of compound concentration (log[I]). Curve-fitting was performed with a dose-response curve from Origin (OriginLab Corporation, Northampton, MA, USA), using the following equation:$$f.a.=\frac{1}{1+10^{\log\left(IC_{50}\right)-\log\left[I\right]}}$$

All data points were collected in triplicate and the final dose-response curves were plotted with MATLAB (R2018a (9.4.0.813654), 2018, MathWorks, Natick, MA, USA). See curves in Appendix A in Appendix A.

### 2.4. Antimalarial Activity

Parasites of the *P. falciparum* strain 3D7 were grown in fresh group 0 positive human erythrocytes, obtained from the Centro Regional de Transfusion Sanguínea-SAS (Granada, Spain) and suspended at 5% haematocrit in RPMI 1640 containing 2% human serum, 0.2% NaHCO_3_, 0.5% Albumax II, 150 μM hypoxanthine, and 12.5 mg/mL gentamicin. Flasks were incubated at 37 °C, under 5% CO_2_ and 95% air mixture. The stock culture was synchronized with 5% sorbitol, and then approximately 96 h later, the parasites were determined to be mostly late ring stages and early trophozoites. The stock culture was then diluted with complete medium and non-parasitized erythrocytes to yield a haematocrit of 2% and parasitemia of 0.25%. Stock solutions of the test drugs were prepared at an initial concentration of 200 mM in DMSO, serially diluted and dispensed into triplicate in 96-well plates to yield final concentrations ranging from 0.0039 to 2 μM. The final well volume was 200 μL with a final concentration of DMSO 0.001%. Each plate also included positive growth controls, where only medium were added, and negative growth controls with 100 nM of chloroquine. (IC_50_ = 0.015 μM). A multidrop dispenser was used to add 50 μL of synchronized parasite culture per well. After 72 h of growth in presence of test compounds, the plates were stored frozen at −20 °C for at least 24 h. Next, they were thawed at room temperature. Parasite growth inhibition assays and IC_50_ determinations were carried out following the *P. falciparum* lactate dehydrogenase assay [9] where parasite viability is quantified by measuring LDH activity upon APAD+ addition, a cofactor that is specific to the Plasmodium enzyme. To evaluate LDH activity, 140 μL of freshly made reaction mix, containing 143 mM sodium L-lactate, 143 μM APAD, 178.75 μM NBT, 1 U/mL diaphorase, 0.7% Tween 20 and 100 mM Tris-HCl (pH 8.0) were dispensed into plates. After 10 min of incubation in the dark at room temperature, absorbance was measured at 630 nm with a vmax precision multiwell plate reader from Molecular Devices, by using the accompanying SOFT maxPro software for analysis. The inhibition percentage was calculated as described [8] and curve fitting was performed using Sigma Plot to yield the drug concentration that produced 50% of the observed decline from the maximum absorbance units in the drug-free control wells (IC_50_). See curves in Appendix A in Appendix A.

## 3. Results and Discussion

### 3.1. Chemistry

#### 3.1.1. General Procedures for the Synthesis of Intermediates Linker and Cationic Heads

The linkers (**a-f**, Figure 1) were synthesized following the procedure previously reported [25,26,27,28]. The cationic heads were obtained following the procedure described [25,26,29]. See Spectra data in Appendix A in Appendix A.

#### 3.1.2. General Procedure for the Synthesis of the Target Compounds

A suspension of 1 equivalent of the linker and 2 equivalents of the corresponding cationic head in CH_3_CN was prepared in a sealed tube. The reaction mixture was stirred at 100 °C for 5 days. Afterwards, the cooled reaction was recrystallized by diethyl ether, filtered to obtain the final product as a solid without purification (Figure 1).

*4-([1,1’-biphenyl]-4-ylmethyl)-7-(pyrrolidin-1-yl)thieno[3,2-b]pyridin-4-ium bromide **FM-a2**.* Following the general procedure, after workup as described previously, compound **FM-a2** was isolated as a light-brown solid. Yield: 82%, mp: 271–273 °C. ^1^H NMR (400 MHz, CD_3_OD) δ 2.14 (dd, *J* = 16.1, 9.0 Hz, 4H), 3.71 (s, 2H), 4.25 (s, 2H), 5.70 (s, 2H), 6.73 (d, *J* = 7.4 Hz, 1H), 7.32 (pt, 1H), 7.32–7.35 (m, 2H), 7.38 (d, *J* = 7.3Hz, 2H), 7.55 (d, *J* = 7.4 Hz, 2H), 7.61 (d, *J* = 8.2 Hz, 2H), 7.64 (d, *J* = 5.8 Hz, 1H), 8.28 (d, *J* = 5.8 Hz, 1H), 8.35 (d, *J* = 7.4 Hz, 1H). ^13^C NMR (101 MHz, CD_3_OD) δ 24.12 (1C), 25.79 (1C), 49.89 (1C), 50.69 (1C), 57.92 (1C), 101.94 (1C), 117.29 (1C), 120.82 (1C), 126.49 (2C), 127.32 (1C), 127.34 (2C), 127.39 (2C), 128.51 (2C), 133.39 (1C), 136.01 (1C), 139.92 (1C), 141.58 (1C), 141.67 (1C), 145.90 (1C), 152.08 (1C). HRMS-*m/z* (M)^+^ calculated for C_24_H_23_N_2_S: 371.1582; found: 371.1585.

*1-([1,1’-biphenyl]-4-ylmethyl)-4-(pyrrolidin-1-yl)thieno[3,2-d]pyrimidin-1-ium bromide **FM-a1**.* Following the general procedure, after workup as described previously, compound **FM-a1** was isolated as a yellow solid. Yield: 69%, mp: 295–298 °C. ^1^H NMR (500 MHz, CD_3_OD) δ 2.09–2.14 (m, 2H), 2.26–2.30 (m, 2H), 3.98–4.04 (m, 2H), 4.19–4.22 (m, 2H), 5.72 (s, 2H), 7.35 (ddd, *J* = 1.5, 2.0, 7.0 Hz, 1H), 7.43 (m, 2H), 7.47 (d, *J* = 8.5 Hz, 2H), 7.59 (d, *J* = 7.3 Hz, 2H), 7.63 (d, *J* = 5.7 Hz, 1H), 7.66 (d, *J* = 8.4 Hz, 2H), 8.45 (d, *J* = 5.7 Hz, 1H), 8.90 (s, 1H). ^13^C NMR (126 MHz, CD_3_OD) δ 23.64 (1C), 25.80 (1C), 49.30 (1C), 50.28 (1C), 55.23 (1C), 116.82 (1C), 117.19 (1C), 126.54 (2C), 127.41 (1C), 127.45 (2C), 127.62 (2C), 128.56 (2C), 132.72 (1C), 138.71 (1C), 139.90 (1C), 141.81 (1C), 146.19 (1C), 149.89 (1C), 155.30 (1C). HRMS-*m/z* (M)^+^ calculated for C_23_H_22_N_3_S: 372.1534; found, 372.1564.

*1-([1,1’-biphenyl]-4-ylmethyl)-4-(pyrrolidin-1-yl)thieno[2,3-d]pyrimidin-1-ium* bromide ***FMa3***. Following the general procedure, after workup as described previously, compound **FMa3** was isolated as a white-grey solid. Yield: 48%, mp: 275–278 °C. ^1^H NMR (400 MHz, DMSO) δ 1.95–197 (m, 2H), 2.05–2.08 (m, 2H), 3.88 (t, *J* = 6.8 Hz, 2H), 4.03 (t, *J* = 6.8 Hz, 2H), 5.70 (s, 2H), 7.34 (t, *J* = 7.3 Hz, 1H), 7.43 (t, *J* = 7.5 Hz, 2H), 7.52 (d, *J* = 8.1, 2H), 7.62 (d, *J* = 7.5 Hz, 2H), 7.69 (d, *J* = 8.2 Hz, 2H), 7.83 (dd, *J* = 26.1, 5.9 Hz, 2H), 9.20 (s, 1H). ^13^C NMR (101 MHz, DMSO) δ 23.91 (1C), 26.24 (1C), 50.72 (1C), 51.15 (1C), 56.96 (1C), 117.39 (1C), 123.85 (1C), 124.76 (1C), 127.14 (2C), 127.61 (2C), 128.25 (1C), 129.25 (2C), 129.42 (2C), 132.22 (1C), 139.62 (1C), 141.15 (1C), 149.62 (1C), 153.39 (1C), 154.39 (1C). HRMS-*m/z* (M)^+^ calculated for C_23_H_22_N_3_S: 372.1534; found: 372.1551.

*1,1’-([1,1’-biphenyl]-4,4’-diylbis(methylene))bis(7-(pyrrolidin-1-yl)thieno[3,2-b]pyridin-4-ium) bromide****Fg-9***. Following the general procedure, after workup as described previously, compound **Fg-9** was isolated as a yellow solid. Yield: 40%, mp: >290 °C. ^1^H NMR (500 MHz, CD_3_OD) δ 2.20 (d, *J* = 18.4 Hz, 8H), 3.75 (s, 4H), 4.28 (s, 4H), 5.73 (s, 4H), 6.76 (d, *J* = 7.4 Hz, 2H), 7.38 (d, *J* = 8.5 Hz, 4H), 7.62 (s, 4H), 7.65 (d, *J* = 5.8 Hz, 2H), 8.30 (d, *J* = 5.8 Hz, 2H), 8.38 (d, *J* = 7.4 Hz, 2H). ^13^C NMR (101 MHz, CD_3_OD) δ 24.27 (4C), 25.81 (4C), 57.85 (2C), 101.96 (2C), 117.26 (2C), 120.81 (2C), 127.33 (4C), 127.46 (4C), 133.98 (2C), 136.02 (2C), 140.45 (2C), 141.70 (2C), 145.88 (2C), 152.09 (2C). HRMS-*m/z* (M)^+^ calculated for C_36_H_36_N_4_S_2_: 588.237; found: 588.2347.

*1,1’-([1,1’-biphenyl]-4,4’-diylbis(methylene))bis(4-(pyrrolidin-1-yl)thieno[3,2-d]pyrimidin-1-ium) bromide **Fa-21.*** Following the general procedure, after workup as described previously, compound **Fa-21** was isolated as a yellow-lemon solid. Yield: 75%, mp: 312–315 °C. ^1^H NMR (500 MHz, CD_3_OD) δ 2.12 (q, *J* = 6.9 Hz, 4H), 2.27 (q, *J* = 6.9 Hz, 4H), 4.01 (t, *J* = 7.0 Hz, 4H), 4.21 (t, *J* = 6.9 Hz, 4H), 5.73 (s, 4H), 7.47 (d, *J* = 8.2 Hz, 4H), 7.61 (d, *J* = 5.7 Hz, 2H), 7.65 (d, *J* = 8.3 Hz, 4H), 8.45 (d, *J* = 5.7 Hz, 2H), 8.91 (s, 2H). ^13^C NMR (126 MHz, CD_3_OD) δ 23.64 (2C), 25.81 (2C), 49.32 (2C), 50.29 (2C), 55.15 (2C), 116.83 (2C), 117.18 (2C), 127.44 (4C), 127.76 (4C), 133.32 (2C), 138.74 (2C), 140.59 (2C), 146.16 (2C), 149.90 (2C), 155.29 (2C). HRMS-*m/z* (M)^2+^ calculated for C_34_H_34_N_6_S_2_: 590.2286; found: 590.2299.

*1,1’-([1,1’-biphenyl]-4,4’-diylbis(methylene))bis(4-(pyrrolidin-1-yl)thieno[2,3-d]pyrimidin-1-ium) bromide **Fg-14**.* Following the general procedure, after workup as described previously, compound **Fg-14** was isolated as a white solid. Yield: 70%, mp: 300–302 °C. ^1^H NMR (400 MHz, DMSO) δ 1.95 (dd, *J* = 13.4, 6.7 Hz, 4H), 2.09 (dt, *J* = 13.8, 6.8 Hz, 4H), 3.87 (t, *J* = 6.7 Hz, 4H), 4.03 (t, *J* = 6.6 Hz, 4H), 5.69 (s, 4H), 7.59 (dd, *J* = 69.1, 8.1 Hz, 8H), 7.83 (dd, *J* = 27.0, 5.9 Hz, 4H), 9.19 (s, 2H). ^13^C NMR (101 MHz, DMSO) δ 154.38 (2C), 153.38 (2C), 149.63 (2C), 140.17 (2C), 132.65 (2C), 129.27 (4C), 124.29 (4C), 124.72 (2C), 123.86 (2C), 117.37 (2C), 56.90 (2C), 51.16 (2C), 50.65 (2C), 26.24 (2C), 23.91 (2C). HRMS-*m/z* (M)^+^ calculated for C_34_H_34_N_6_S_2_: 590.2275; found: 590.2261.

*1,1’-([1,1’-biphenyl]-4,4’-diylbis(methylene))bis(4-(piperidin-1-yl)thieno[3,2-d]pyrimidin-1-ium) bromide****Fa-24**.* Following the general procedure, after workup as described previously, compound **Fa-24** was isolated as a yellow-lemon solid. Yield: 80.36%, mp: 312–315 °C. ^1^H NMR (400 MHz, CD_3_OD) δ 1.83–186 (m, 12H), 4.24 (d, *J* = 19.3 Hz, 8H), 5.72 (s, 4H), 7.45 (d, *J* = 8.1 Hz, 4H), 7.61 (d, *J* = 5.9 Hz, 2H), 7.63 (d, *J* = 8.2 Hz, 4H), 8.40 (d, *J* = 5.8 Hz, 2H), 8.90 (s, 2H). ^13^C NMR (101 MHz, CD_3_OD) δ 23.44 (2C), 25.61 (2C), 26.34 (2C), 47.77 (2C), 50.11 (2C), 55.22 (2C), 115.04 (2C), 117.47 (2C), 127.44 (4C), 127.74 (4C), 133.24 (2C), 138.09 (2C), 140.60 (2C), 147.50 (2C), 149.51 (2C), 155.82 (2C). HRMS–*m/z* (M)^+^ calculated for C_36_H_38_N_6_S_2_: 618.2599, found: 618.2609.

*1,1’-([1,1’-biphenyl]-4,4’-diylbis(methylene))bis(4-(piperidin-1-yl)thieno[2,3-d]pyrimidin-1-ium) bromide **Fg-30.**
*Following the general procedure, after workup as described previously, compound **Fg-30** was isolated as a white solid. Yield: 57%, mp: 388–290 °C. ^1^H NMR (500 MHz, CD_3_OD) δ 1.95–1.97 (m, 12H), 4.17 (s, 4H), 4.29 (s, 4H), 5.70 (s, 4H), 7.63 (d, *J* = 8.3 Hz, 8H), 7.78 (d, *J* = 6.0 Hz, 4H), 8.93 (s, 2H). ^13^C NMR (126 MHz, CD_3_OD) δ 23.39 (6C), 50.35 (2C), 56.99 (4C), 116.26 (2C), 123.04 (2C), 124.42 (2C), 127.49 (4C), 128.69 (4C), 131.62 (2C), 141.02 (2C), 148.22 (2C), 154.83 (2C), 155.99 (2C). HRMS-*m/z* (M)^+^ calculated for C_36_H_38_N_8_S_2_: 618.2588, found: 618.2568.

*1,1’-([1,1’-biphenyl]-4,4’-diylbis(methylene))bis(7-(azepan-1-yl)thieno[3,2-b]pyridin-4-ium) bromide **Fg-10.**
*Following the general procedure, after workup as described previously, compound **Fg-10** was isolated as a white solid. Yield: 53%, mp: 220–222 °C. ^1^H NMR (600 MHz, CD_3_OD) δ 1.68 (s, 8H), 2.01 (s, 8H), 4,08 (s, 8H), 5.76 (s, 4H), 6.98 (d, *J* = 7.5 Hz, 2H), 7.38 (d, *J* = 8.1 Hz, 2H), 7.48 (d, *J* = 8.1 Hz, 2H), 7.57 (d, *J* = 8.2 Hz, 2H), 7.65 (d, *J* = 8.2 Hz, 2H), 7.68 (d, *J* = 5.9 Hz, 2H), 8.31 (d, *J* = 5.9 Hz, 2H), 8.40 (d, *J* = 7.5 Hz, 2H). ^13^C NMR (101 MHz, CD_3_OD) δ 26.11 (4C), 32.23 (4C), 52.40 (4C), 58.08 (2C), 101.75 (2C), 117.41 (2C), 119.20 (2C), 126.62 (2C), 127.39 (2C), 127.50 (2C), 129.37 (2C), 133.64 (2C), 135.90 (2C), 137.88 (2C), 139.89 (2C), 140.90 (2C), 141,71 (2C), 146.55 (2C), 154,18 (2C). HRMS-*m/z* (M)^+^ calculated for C_40_H_44_N_4_S_2_: 644.2996; found: 644.3033.

*1,1’-([1,1’-biphenyl]-4,4’-diylbis(methylene))bis(4-(azepan-1-yl) thieno[2,3-d]pyrimidin-1-ium) bromide **Fa-22.**
*Following the general procedure, after workup as described previously, compound **Fa-22** was isolated as a yellow-brown solid. Yield: 47.22%, mp: 304–306 °C. ^1^H NMR (500 MHz, CD_3_OD) δ 1.69–171 (m, 8H), 1.96–197 (m, 4H), 2.06–2.08 (m, 4H), 4.21–4.23 (m, 4H), 4.24–4.27 (m, 4H), 5.74 (s, 4H), 7.48 (d, *J* = 8.3 Hz, 4H), 7.63 (d, *J* = 5.8 Hz, 2H), 7.67 (d, *J* = 8.3 Hz, 4H), 8.46 (d, *J* = 5.8 Hz, 2H), 8.92 (s, 2H). ^13^C NMR (126 MHz, CD_3_OD) δ 26.08 (2C), 26.20 (2C), 26.31 (2C), 28.09 (2C), 50.34 (2C), 51.35 (2C), 55.24 (2C), 115.28 (2C), 117.28 (2C), 127.49 (4C), 127.77 (4C), 133.23 (2C), 138.73 (2C), 140.66 (2C), 147.05 (2C), 149.45 (2C), 157.11 (2C). HRMS-*m/z* (M)^+^ calculated for C_38_H_42_N_6_S_2_: 646.2912, found: 646.2885. 

*1,1’-([1,1’-biphenyl]-4,4’-diylbis(methylene))bis(4-(azepan-1-yl)thieno[3,2-d]pyrimidin-1-ium) bromide **Fg-18**.* Following the general procedure, after workup as described previously, compound **Fg-18** was isolated as a white solid. Yield: 74%, mp: 303–305 °C. ^1^H NMR (400 MHz, CD_3_OD) δ 1.68–1.66 (m, 8H), 1.94 (s, 4H), 2.01 (s, 4H), 4.16–4.12 (m, 4H), 4.21–4.17 (m, 4H), 5.68 (s, 4H), 7.61 (d*, J =* 8.3 Hz, 8H), 7.76 (d, *J* = 6.0 Hz, 4H), 8.94 (s, 2H). ^13^C NMR (101 MHz, CD_3_OD) δ 26.16 (2C), 26.37 (2C), 26.46 (2C), 27.35 (2C), 50.33 (2C), 51.56 (2C), 57.03 (2C), 116.37 (2C), 123.31 (2C), 123.67 (2C), 127.46 (4C), 128.69 (4C), 131.53 (2C), 140.9 (2C), 148.05 (2C), 154.54 (2C), 156.60 (2C). HRMS-*m/z* (M)^+^ calculated for C_38_H_42_N_6_S_2_: 646.2901; found: 646.2908.

*1,1’-([1,1’-biphenyl]-4,4’-diylbis(methylene))bis(4-(methyl(phenyl)amino)thieno[3,2-d]pyrimidin-1-ium)bromide **Fp-1.**
*Following the general procedure, after workup as described previously, compound **Fp-1** was isolated as a blue solid. Yield: 20%, mp: 185–187 °C. ^1^H NMR (400 MHz, CD_3_OD) δ 3.84 (s, 6H). 5.80 (s, 4H), 7.71–7.44 (m, 20H), 8.14 (d, *J* = 5.7 Hz, 2H), 9.16 (s, 2H). ^13^C NMR (101 MHz, CD_3_OD) δ 41.78 (2C), 57.05 (2C), 117.92 (2C), 128.23 (4C), 128.87 (4C), 129.20 (4C), 130.36 (4C), 131.84 (2C), 134.26 (2C), 138.96 (2C), 141.25 (2C), 141.77 (2C), 142.52 (2C), 148.62 (2C), 151.66 (2C), 159.00 (2C). HRMS-*m/z* (M)^+^ calculated for C_40_H_34_N_6_S_2_: 662.2286, found: 662.2267.

1,1’-([1,1’-biphenyl]-4,4’-diylbis(methylene))bis(4-((4-chlorophenyl)(methyl)amino)thieno[2,3-d]pyrimidin-1-ium) bromide ***Fp-8.*** Following the general procedure, after workup as described previously, compound **Fp-8** was isolated as a white solid. Yield: 40%, mp: 264–265 °C. ^1^H NMR (400 MHz, DMSO) δ 3.76 (s, 6H), 5.81 (s, 4H), 7.74–7.57 (m, 20H), 9.44 (s, 2H). ^13^C NMR (126 MHz, DMSO) δ 41.85 (2C), 56.94 (2C), 117.33 (4C), 125.60 (2C), 127.27 (6C), 129.00 (8C), 130.74 (4C), 132.00 (2C), 139.82 (4C), 149.66 (2C), 154.25 (2C), 156.60 (2C). HRMS-*m/z* (M)^+^ calculated for C_40_H_32_N_6_S_2_Cl_2_: 730.1507, found: 730.1520.

*4,4’-([2,2’-bipypidine]-5,5’-diylbis(methylene))bis(7-(pyrrolidin-1-yl)thieno[3,2-b]pyridin-4-ium bromide).**Fg-12**.* Following the general procedure, after workup as described previously, compound **Fg-12** was isolated as a purple solid. Yield: 60%, mp: >290 °C. ^1^H NMR (400 MHz, DMSO) δ 2.05 (s, 4H), 3.63 (s, 4H), 4.09 (d, *J* = 28.6 Hz, 8H), 5.83 (s, 4H), 6.80 (d, *J* = 7.4 Hz, 2H), 7.78 (dd, *J* = 8.4, 2.0 Hz, 2H), 7.82 (d, *J* = 5.8 Hz, 2H), 8.28 (d, *J* = 8.3 Hz, 2H), 8.49 (d, *J* = 5.8 Hz, 2H), 8.61 (d, *J* = 7.4 Hz, 2H), 8.69 (d, *J* = 1.4 Hz, 2H). ^13^C NMR (101 MHz, DMSO) δ 24.41 (2C), 26.36 (2C), 51.04 (2C), 51.47 (2C), 65.33 (2C) 102.99 (2C), 118.15 (2C), 120.55 (2C), 120.98 (2C), 132.09 (2C), 136.65 (2C), 137.66 (2C), 142.59 (2C), 145.78 (2C), 148.85 (2C), 151.85 (2C), 155.07 (2C). HRMS-*m/z* (M)^+^ calculated for C_34_H_34_N_6_S_2_: 590.2275; found: 590.2247.

*1,1’-([2,2’-bipyridine]-5,5’-diylbis(methylene))bis(4-(pyrrolidin-1-yl)thieno[3,2-d]pyrimidin-1-ium) bromide* ***Fg-17.*** Following the general procedure, after workup as described previously, compound **Fg-17** was isolated as a yellow solid. Yield: 62%, mp: 330–332 °C. ^1^H NMR (400 MHz, CD_3_OD) δ 2.09 (q, *J* = 6.8 Hz, 4H), 2.25 (q, *J* = 6.8 Hz, 4H), 3.99 (t, *J* = 6.9 Hz, 4H), 4.19 (t, *J* = 6.9 Hz, 4H), 5.79 (s, 4H), 7.63 (d, *J* = 5.7 Hz, 2H), 7.90 (dd, *J* = 8.3, 2.0 Hz, 2H), 8.35 (d, *J* = 8.3 Hz, 2H), 8.45 (d, *J* = 5.7 Hz, 2H), 8.72 (d, *J* = 1.5 Hz, 2H), 8.91 (s, 2H). ^13^C NMR (101 MHz, CD_3_OD) δ 23.61 (2C), 25.77 (2C), 49.35 (2C), 50.33 (2C), 52.77 (2C), 116.87 (2C), 121.13 (2C), 130.57 (2C), 136.20 (4C), 138.98 (2C), 145.89 (2C), 148.16 (2C), 149.96 (2C), 155.29 (2C), 155.35 (2C). HRMS-*m/z* (M)^+^ calculated for C_32_H_32_N_8_S_2_: 592.2180; found: 592.2235.

*1,1’-([2,2’-bipyridine]-5,5’-diylbis(methylene))bis(4-(pyrrolidin-1-yl)thieno[2,3-d]pyrimidin-1-ium) bromide **Fg-13.*** Following the general procedure, after workup as described previously, compound **Fg-13** was isolated as a white solid. Yield: 71%, mp: 238–240 °C. ^1^H NMR (400 MHz, DMSO) δ 2.14–1.91 (m, 8H), 3.87 (t, *J* = 6.9 Hz, 4H), 4.03 (t, *J* = 6.9 Hz, 4H), 5.79 (s, 2H), 7.80 (d, *J* = 5.9 Hz, 2H), 7.87 (d, *J* = 5.9 Hz, 2H), 8.01 (dd, *J* = 8.3, 2.3 Hz, 2H), 8.35 (d, *J* = 8.3 Hz, 2H), 8.83 (d, *J* = 2.3 Hz, 2H), 9.21 (s, 2H). ^13^C NMR (101 MHz, DMSO) δ 23.91 (2C), 26.24 (2C), 50.75 (2C), 51.18 (2C), 54.67 (2C), 117.43 (2C), 121.04 (2C), 123.96 (2C), 124.62 (2C), 129.78 (2C), 137.72 (2C), 149.75 (2C), 155.44 (2C), 149.86 (2C), 153.25 (2C), 154.41 (2C). HRMS-m/z (M)^+^ calculated for C_32_H_32_N_8_S_2_: 592.2180; found: 592.2200. 

*1,1’-([2,2’-bipyridine]-5,5’-diylbis(methylene))bis(4-(piperidin-1-yl)thieno[3,2-d]pyrimidin-1-ium) bromide **Fa-27**.* Following the general procedure, after workup as described previously, compound **Fa-27** was isolated as a yellow-brown solid. Yield: 58%, mp: 307–309 °C. ^1^H NMR (500 MHz, CD_3_OD) δ 1.86–189 (m, 12H), 4.29–4.27 (m, 8H), 5.84 (s, 4H), 7.68 (d, *J* = 5.8 Hz, 2H), 7.93 (dd, *J* = 8.3, 2.2 Hz, 2H), 8.38 (d, *J* = 8.3 Hz, 2H), 8.46 (d, *J* = 5.8 Hz, 2H), 8.76 (d, *J* = 2.0 Hz, 2H), 8.96 (s, 2H). ^13^C NMR (126 MHz, CD_3_OD) δ 23.45 (2C), 25.62 (2C), 26.35 (2C), 47.81 (2C), 50.19 (2C), 52.90 (2C), 115.16 (2C), 117.24 (2C), 121.18 (2C), 130.55 (2C), 136.28 (2C), 138.38 (2C), 147.30 (2C), 148.22 (2C), 149.62 (2C), 155.50 (2C), 155.85 (2C). HRMS-*m/z* (M)^+^ calculated for C_34_H_36_N_8_S_2_: 620.2504, found: 620.2446.

*1,1’-([2,2’-bipyridine]-5,5’-diylbis(methylene))bis(4-(piperidin-1-yl)thieno[2,3-d]pyrimidin-1-ium) bromide **Fg-32**.* Following the general procedure, after workup as described previously, compound **Fg-32** was isolated as a white solid. Yield: 64%, mp: 267–269C°. ^1^H NMR (400 MHz, CD_3_OD) δ 8.94 (s, 2H), 8.81 (s, 2H), 8.41 (d, *J* = 8.2 Hz, 2H), 8.01 (d, *J* = 8.2 Hz, 2H), 7.76 (d, *J* 5.9 Hz, 4H), 5.76 (s, 4H), 4.27 (s, 4H), 4.14 (s, 4H), 1.84 (s, 12H). ^13^C NMR (101 MHz, CD_3_OD) δ 155.95 (2C), 155.80 (2C), 154.55 (2C), 149.07 (2C), 148.31 (2C), 137.05 (2C), 129.01 (2C), 123.36 (2C), 123.23 (2C), 121.20 (2C), 116.29 (2C), 54.53 (2C), 50.23 (2C), 24.93 (2C), 23.34 (6C). HRMS-*m/z* (M)^+^ calculated for C_34_H_36_N_8_S_2_: 620.2493; found: 620.2459.

*1,1’-([2,2’-bipyridine]-5,5’-diylbis(methylene))bis(4-(azepan-1-yl)thieno[3,2-d]pyrimidin-1-ium) bromide **Fa-26**.* Following the general procedure, after workup as described previously, compound **Fa-26** was isolated as a light brown solid. Yield: 20%, mp: 310–312 °C. ^1^H NMR (400 MHz, CD_3_OD) δ 1.67–1.69 (m, 8H), 1.96–198 (m, 4H), 2.02–2.04 (m, 4H), 4.20–4.22 (m, 8H), 5.80 (s, 4H), 7.64 (d, *J* = 5.8 Hz, 2H), 7.91 (d, *J* = 8.0 Hz, 2H), 8.36 (d, *J* = 8.3 Hz, 2H), 8.46 (d, *J* = 5.8 Hz, 2H), 8.73 (d, *J* = 1.3 Hz, 2H), 8.92 (s, 2H). ^13^C NMR (126 MHz, CD_3_OD) δ 26.07 (2C), 26.19 (2C), 26.31 (2C), 28.04 (2C), 50.38 (2C), 51.40 (2C), 52.89 (2C), 115.39 (2C), 117.02 (2C), 121.18 (2C), 130.51 (2C), 136.29 (2C), 139.00 (2C), 146.82 (2C), 148.23 (2C), 149.54 (2C), 155.51 (2C), 157.12 (2C). HRMS–*m/z* (M)^+^ calculated for C_36_H_40_N_8_S_2_: 648.2817, found: 648.2833.

*1,1’-([2,2’-bipyridine]-5,5’-diylbis(methylene))bis(4-(azepan-1-yl)thieno[2,3-d]pyrimidin-1-ium) bromide **Fg-20**.* Following the general procedure, after workup as described previously, compound **Fg-20** was isolated as a white solid. Yield: 68%, mp: 289–291 °C. ^1^H NMR (500 MHz, CD_3_OD) δ 1.74–1.66 (m, 8H), 2.00–1.94 (m, 4H), 2.08–2.01 (m, 4H), 4.19–4.16 (m, 4H), 4.24–4.21 (m, 4H), 5.81 (s, 4H), 7.75 (d, *J* = 6.0 Hz, 2H), 7.84 (d, *J* = 6.0 Hz, 2H), 8.04 (dd, *J* = 8.3, 2.3 Hz, 2H), 8.44 (d, *J* = 8.3 Hz, 2H), 8.84 (d, *J* = 2.1 Hz, 2H), 8.96 (s, 2H). ^13^C NMR (126 MHz, CD_3_OD) δ 26.40 (4C), 26.48 (2C), 27.35 (2C), 50.39 (2C), 51.63 (2C), 54.63 (2C), 116.48 (2C), 121.24 (2C), 123.16 (2C), 123.92 (2C), 128.97 (2C), 137.10 (2C), 148.20 (2C), 149.12 (2C), 154.33 (2C), 155.83 (2C), 156.64 (2C). HRMS-*m/z* (M)^+^ calculated for C_36_H_40_N_8_S_2_: 648.2806; found: 648.2794.

*4,4’-((ethane-1,2-diylbis(4,1-phenylene))bis(methylene))bis(7-(pyrrolidin-1-yl)thieno[3,2-b]pyridin-4-ium) bromide **Fg-11***. Following the general procedure, after workup as described previously, compound **Fg-11** was isolated as a white solid. Yield: 68%, mp: 253–255 °C. ^1^H NMR (400 MHz, CD_3_OD) δ 2.17 (s, 8H), 3.29 (dt, *J* = 3.2, 1.6 Hz, 2H), 3.33 (s, 2H), 3,71 (s, 4H), 4.25 (s, 4H), 5.61 (s, 4H), 6.71 (d, *J* = 7.4 Hz, 2H), 7.15 (s, 8H), 7.58 (d, *J* = 5.8 Hz, 2H), 8.27 (d, *J* = 5.8 Hz, 2H), 8.29 (d, *J* = 7.4 Hz, 2H). ^13^C NMR (126 MHz, CD_3_OD) δ 24.05 (4C), 25.69 (4C), 36.88 (1C), 50.71 (1C), 58.04 (2C), 101.90 (2C), 117.29 (2C), 120.81 (2C), 126.92 (4C), 129.00 (4C), 132.08 (2C), 135.92 (2C), 141.63 (2C), 141.96 (2C), 145.91 (2C), 152.08 (2C). HRMS-*m/z* (M)^+^ calculated for C_38_H_40_N_4_S_2_: 616.2683; found: 616.2717.

*1,1’-((ethane-1,2-diylbis(4,1-phenylene))bis(methylene))bis(4-(pyrrolidin-1-yl)thieno[3,2-d]pyrimidin-1-ium) bromide **Fg-16***. Following the general procedure, after workup as described previously, compound **Fg-16** was isolated as a yellow solid. Yield: 72%, mp: 231–233 °C. ^1^H NMR (400 MHz, CD_3_OD) δ 2.09 (d, *J* = 6.8 Hz, 4H), 2.24 (d, *J* = 6.8 Hz, 4H), 2.85 (s, 2H), 3.29 (dt, *J* = 3.0, 1.5 Hz, 2H), 3.98 (t, *J* = 7.0 Hz, 4H), 4.18 (t, *J* = 6.9 Hz, 4H), 5.60 (s, 4H), 7.20 (d, *J* = 8.1 Hz, 4H), 7,24 (d, *J* = 8,1 Hz, 4H), 7.54 (d, *J* = 5.7 Hz, 2H), 8.42 (d, *J* = 5.7 Hz, 2H), 8.81 (s, 2H). ^13^C NMR (101 MHz, CD_3_OD) δ 23.61 (4C), 25.78 (2C), 36.81 (2C), 49.28 (1C), 50.23 (1C), 55.31 (2C), 116.77 (2C), 117.16 (2C), 127.15 (4C), 129.09 (4C), 131.31 (2C), 138.62 (2C), 142.42 (2C), 146.16 (2C), 149.75 (2C), 155.25 (2C). HRMS-*m/z* (M)^+^ calculated for C_36_H_38_N_6_S_2_: 618.2588; found: 618.2586.

*1,1’-((ethane-1,2-diylbis(4,1-phenylene))bis(methylene))bis(4-(pyrrolidin-1-yl)thieno[2,3-d]pyrimidin-1-ium) bromide **Fg-15**.* Following the general procedure, after workup as described previously, compound **Fg-15** was isolated as a white solid. Yield: 80%, mp: 293–295 °C. ^1^H NMR (400 MHz, CD_3_OD) δ 2.10 (q, *J* = 6.9 Hz, 2H), 2.23 (q, *J* = 6.9 Hz, 2H), 2.88 (s, 4H), 4.00 (t, *J* = 6.9 Hz, 4H), 4.11 (t, *J* = 6.9 Hz, 4H), 5.56 (s, 4H), 7.19 (d, *J* = 8.0 Hz, 4H), 7.30 (d, *J* = 8.0 Hz, 4H), 7.79 (d, *J* = 6.0Hz, 2H), 7.88 (d, *J* = 6.0 Hz, 2H), 8.86 (s, 2H). ^13^C NMR (126 MHz, CD_3_OD) δ 23.43 (4C), 25.88 (2C), 36.90 (2C), 50.33 (1C), 50.78 (1C), 57.19 (2C), 117.34 (2C), 123.08 (2C), 123.29 (2C), 128.06 (4C), 129.16 (4C), 129.57 (2C), 142.98 (2C), 148.42 (2C), 153.48 (2C), 154.51 (2C). HRMS-*m/z* (M)^+^ calculated for C_36_H_38_N_6_S_2_: 618.2588; found: 618.2586.

*1,1’-((ethane-1,2-diylbis(4,1-phenylene))bis(methylene))bis(4-(piperidin-1-yl)thieno[3,2-d]pyrimidin-1-ium) bromide **Fa-25**.* Following the general procedure, after workup as described previously, compound **Fa-25** was isolated as a yellow-brown solid. Yield: 76%, mp: 314–316 °C. ^1^H NMR (400 MHz, CD_3_OD) δ 1.82–1.85 (m, 12H), 2.84–2.87 (s, 4H), 4.24–4.27 (m, 8H), 5.62 (s, 4H), 7.19 (d, *J* = 8.1 Hz, 4H), 7.25 (d, *J* = 8.1 Hz, 4H), 7.57 (d, *J* = 5.8 Hz, 2H), 8.40 (d, *J* = 5.7 Hz, 2H), 8.83 (s, 2H). ^13^C NMR (101 MHz, CD_3_OD) δ 23.48 (2C), 25.70 (2C), 26.42 (2C), 36.88 (2C), 47.75 (2C), 50.11 (2C), 55.52 (2C), 114.99 (2C), 117.56 (2C), 127.27 (4C), 129.18 (4C), 131.10 (2C), 138.17 (2C), 142.50 (2C), 147.55 (2C), 149.34 (2C), 155.78 (2C). HRMS-*m/z* (M)^+^ calculated for C_38_H_42_N_6_S_2_: 646.2912, found: 646.2922.

*1,1’-((ethane-1,2-diylbis(4,1-phenylene))bis(methylene))bis(4-(piperidin-1-yl)thieno[2,3-d]pyrimidin-1-ium) bromide **Fg-31**.* Following the general procedure, after workup as described previously, compound **Fg-31** was isolated as a white solid. Yield: 85%, mp: 295–297 °C. ^1^H NMR (400 MHz, CD_3_OD) δ 1.83–1.85 (s, 12H), 2.88 (s, 2H), 3.29 (s, 2H), 4.13 (s, 4H), 4.25 (s, 4H), 5.56 (s, 4H), 7.32 (dd, *J* = 7.9 Hz, 8H), 7.75 (dd, *J* = 19.1, 5.9 Hz, 4H), 8.84 (s, 2H). ^13^C NMR (101 MHz, CD_3_OD) δ 23.36 (6C), 25.90 (2C), 36.87 (2C), 50.17 (2C) 57.17 (2C), 116.22 (2C), 122.95 (2C), 123.57 (2C), 128.10 (4C), 129.14 (4C), 129.53 (2C), 142.97 (2C), 148.07 (2C), 154.79 (2C), 155.95 (2C). HRMS-*m/z* (M)^+^ calculated for C_38_H_42_N_6_S_2_: 646.2901; found: 646.2946.

*1,1’-((ethane-1,2-diylbis(4,1-phenylene))bis(methylene))bis(4-(azepan-1-yl)thieno[3,2-d]pyrimidin-1-ium) bromide **Fa-23***. Following the general procedure, after workup as described previously, compound **Fa-23** was isolated as a yellow-orange solid. Yield: 81%, mp: 321–323 °C. ^1^H NMR (500 MHz, CD_3_OD) δ 1.67–1.70 (m, 8H), 1.95–1.98 (m, 4H), 2.05–2.07 (m, 4H), 2.89 (s, 4H), 4.21 (t, *J* = 6 Hz, 4H), 4.25 (t, *J* = 6 Hz, 4H), 5.65 (s, 4H), 7.22 (d, *J* = 8.2 Hz, 4H), 7.29 (d, *J* = 8.2 Hz, 4H), 7.59 (d, *J* = 5.8 Hz, 2H), 8.47 (t, *J* = 5.9 Hz, 2H), 8.87 (s, 2H). ^13^C NMR (126 MHz, CD_3_OD) δ 26.09 (2C), 26.21 (2C), 26.32 (2C), 28.11 (2C), 36.86 (2C), 50.31 (2C), 51.34 (2C), 55.44 (2C), 115.25 (2C), 117.33 (2C), 127.24 (4C), 129.14 (4C), 131.26 (2C), 138.66 (2C), 142.52 (2C), 147.07 (2C), 149.34 (2C), 157.09 (2C). HRMS-*m/z* (M)^+^ calculated for C_40_H_46_N_6_S_2_: 674.3225, found: 674.3286.

*1,1’-((ethane-1,2-diylbis(4,1-phenylene))bis(methylene))bis(4-(azepan-1-yl)thieno[2,3-d]pyrimidin-1-ium) bromide **Fg-19***. Following the general procedure, after workup as described previously, compound **Fg-19** was isolated as a white solid. Yield: 78%, mp: 298–300 °C. ^1^H NMR (500 MHz, CD_3_OD) δ 1.69–1.71 (m, 8H), 2.02–2.06 (s, 8H), 2.92 (s, 4H), 4.18–4.15 (m, 4H), 4.23–4.19 (m, 4H), 5.60 (s, 4H), 7.24 (d, *J* = 8.1 Hz, 4H), 7.35 (d, *J* = 8.1 Hz, 4H), 7.79 (d, *J* = 6.0 Hz, 2H), 7.81 (d, 6.0 Hz, 2H), 8.89 (s, 2H). ^13^C NMR (101 MHz, CD_3_OD) δ 26.18 (2C), 26.39 (2C), 26.48 (2C), 27.38 (2C), 36.92 (2C), 50.34 (2C), 51.56 (2C), 57.24 (2C), 116.38 (2C), 123.33 (2C), 123.64 (2C), 128.13 (4C), 129.16 (4C), 129.49 (2C), 143.04 (2C), 147.96 (2C), 154.56 (2C), 156.62 (2C). HRMS-*m/z* (M)^+^ calculated for C_40_H_46_N_6_S_2_: 646.3214; found: 646.3174.

*4,4’-((butane-1,4-diylbis(4,1-phenylene))bis(methylene))bis(7-(pyrrolidin-1-yl)thieno[3,2-b]pyridin-4-ium) bromide****Ff-1***. Following the general procedure, after workup as described previously, compound **Ff-1** was isolated as a white solid. Yield: 17%, mp: 195–197 °C. ^1^H NMR (400 MHz, CD_3_OD) δ 1.61 (s, 4H), 2.14–2.16 (m, 8H), 2.62 (s, 4H), 3.75–3.78 (m, 4H), 4.29–4.32 (m, 4H), 5.67 (s, 4H), 6.77 (d, *J* = 7.4 Hz, 2H), 7.20 (d, *J* = 8.3 Hz, 4H), 7.23 (d, *J* = 8.3 Hz, 4H), 7.66 (d, *J* = 5.8 Hz, 2H), 8.32 (d, *J* = 5.8 Hz, 2H), 8.36 (d, *J* = 7.4 Hz, 2H). ^13^C NMR (101 MHz, CD_3_OD) δ 25.11(2C), 26.80 (2C), 31.60 (2C), 35.79 (2C), 51.49 (2C), 51.70 (2C), 59.04 (2C), 102.91 (2C), 118.30 (2C), 121.77 (2C), 127.95 (4C), 129.85 (4C), 132.79 (2C), 136.92 (2C), 142.60 (2C), 144.24 (2C), 146.86 (2C), 153.06 (2C). HRMS-*m/z* (M)^+^ calculated for C_40_H_44_N_4_S_2_: 644.2996; found: 648.2584.

*1,1’-((butane-1,4-diylbis(4,1-phenylene))bis(methylene))bis(4-(pyrrolidin-1-yl)thieno[3,2-d]pyrimidin-1-ium) bromide **Ff-7***. Following the general procedure, after workup as described previously, compound **Ff-7** was isolated as a yellow solid. Yield: 19%, mp: 215–217 °C. ^1^H NMR (400 MHz, CD_3_OD) δ 1.62 (s, 4H), 2.13–2.15 (m, 4H), 2.29–2.32 (m, 4H), 2.64 (s, 4H), 4.01–4.04 (m, 4H), 4.21–4.23 (m, 4H), 5.66 (s, 4H), 7.22 (d, *J* = 8.0 Hz, 4H), 7.32 (d, *J* = 8.0 Hz, 4H), 7.62 (d, *J* = 5.7 Hz, 2H), 8.46 (d, *J* = 5.7 Hz, 2H), 8.88 (s, 2H). ^13^C NMR (101 MHz, CD_3_OD) δ 24.41 (2C), 26.58 (2C), 31.37 (2C), 35.58 (2C), 50.06 (2C), 51.02 (2C), 56.12 (2C), 117.56 (2C), 117.96 (2C), 127.95 (4C), 129.73 (4C), 131.85 (2C), 139.41 (2C), 144.25 (2C), 146.96 (2C), 150.55 (2C), 156.05 (2C). HRMS-*m/z* (M)^+^ calculated for C_38_H_42_N_6_S_2_: 646.2901; found: 646.2936.

*1,1’-((butane-1,4-diylbis(4,1-phenylene))bis(methylene))bis(4-(pyrrolidin-1-yl)thieno[2,3-d]pyrimidin-1-ium) bromide **Ff-3***. Following the general procedure, after workup as described previously, compound **Ff-3** was isolated as a grey solid. Yield: 42%, mp: 288–290 °C. ^1^H NMR (500 MHz, CD_3_OD) δ 1.63 (s, 4H), 2.12–2.14 (m, 4H), 2.26–2.28 (m, 4H), 2.66 (s, 4H), 4.03–4.06 (m, 4H), 4.14–4.17 (m, 4H), 5.62 (s, 4H), 7.25 (d, *J* = 8.1 Hz, 4H), 7.37 (d, *J* = 8.1 Hz, 4H), 7.73 (d, *J* = 5.9 Hz, 2H), 7.92 (d, *J* = 6.0 Hz, 2H), 8.92 (s, 2H). ^13^C NMR (126 MHz, CD_3_OD) δ 24.33 (2C), 26.78 (2C), 31.50 (2C), 35.76 (2C), 51.22 (2C), 51.67 (2C), 58.12 (2C), 118.25 (2C), 123.98 (2C), 124.17 (2C), 128.97 (4C), 129.87 (4C), 130.22 (2C), 144.97 (2C), 149.32 (2C), 154.40 (2C), 155.42 (2C). HRMS*-m/z* (M)^+^ calculated for C_38_H_42_N_6_S_2_: 646.2901; found: 646.2930.

*1,1’-((butane-1,4-diylbis(4,1-phenylene))bis(methylene))bis(4-(piperidin-1-yl)thieno[3,2-d]pyrimidin-1-ium) bromide ****Fa-33***. Following the general procedure, after workup as described previously, compound **Fa-33** was isolated as a light-yellow solid. Yield: 89%, mp: 276–278 °C. ^1^H NMR (400 MHz, CD_3_OD) δ 1.57 (m, 4H), 1.82–1.84 (m, 12H), 2.59 (t, *J* = 6.7 Hz, 4H), 4.21–4.24 (m, 8H), 5.62 (s, 4H), 7.18 (d, *J* = 8.2 Hz, 4H), 7.26 (d, *J* = 8.2 Hz, 4H), 7.58 (d, *J* = 5.8 Hz, 2H), 8.39 (d, *J* = 5.8 Hz, 2H), 8.84 (s, 2H). ^13^C NMR (101 MHz, CD_3_OD) δ 155.82 (2C), 149.38 (2C), 147.54 (2C), 143.51 (2C), 137.97 (2C), 130.99 (2C), 128.95 (4C), 127.15 (4C), 117.47 (2C), 115.00 (2C), 55.40 (2C), 50.09 (4C), 34.79 (2C), 30.61 (2C), 25.76 (4C), 23.44 (2C). HRMS-*m/z* (M)^+^ calculated for C_40_H_46_N_6_S_2_: 674.3225, found: 674.3190.

*1,1’-((butane-1,4-diylbis(4,1-phenylene))bis(methylene))bis(4-(piperidin-1-yl)thieno[2,3-d]pyrimidin-1-ium) bromide ****Ff-6***. Following the general procedure, after workup as described previously, compound **Ff-6** was isolated as a grey solid. Yield: 11%, mp: 282–284 °C. ^1^H NMR (400 MHz, CD_3_OD) δ 1.66 (s, 4H), 1.90–1.92 (m, 12H), 2.68 (s, 4H), 4.30–4.32 (m, 8H), 5.63 (s, 4H), 7.27 (d, *J* = 8.0 Hz, 4H), 7.40 (d, *J* = 8.5 Hz, 4H), 7.78 (d, *J* = 5.8 Hz, 2H), 7.84 (d, *J* = 6.2 Hz, 2H), 8.91 (s, 2H). ^13^C NMR (126 MHz, CD_3_OD) δ 24.23 (4C), 26.39 (2C), 31.44 (2C), 35.70 (2C), 51.05 (4C), 58.06 (2C), 117.08 (2C), 123.81 (2C), 124.42 (2C), 128.97 (4C), 129.82 (4C), 130.89 (2C), 144.92 (2C), 148.92 (2C), 155.67 (2C), 156.82 (2C). HRMS-*m/z* (M)^+^ calculated for C_40_H_46_N_6_S_2_: 674.3225, found: 674.3239.

*(1,1’((butane1,4diylbis(4,1phenylene))bis(methylene))bis(4-(azepan-1-yl)thieno[3,2-d]pyrimidin-1-ium)) bromide ****Fa-29***. Following the general procedure, after workup as described previously, compound **Fa-29** was isolated as a light-brown solid. Yield: 40%, mp: 268–270 °C. ^1^H NMR (500 MHz, CD_3_OD) δ 1.59–1.61 (m, 4H), 1.69–1.71 (m, 8H), 1.96–1.98 (m, 4H), 2.04–2.06 (m, 4H), 2.62 (t, *J* = 6.9 Hz, 4H), 4.20–4.22 (m, 4H), 4.24–4.26 (m, 4H), 5.65 (s, 4H), 7.21 (d, *J* = 7.9 Hz, 4H), 7.30 (d, *J* = 7.9 Hz, 4H), 7.61 (d, *J* = 5.8 Hz, 2H), 8.45 (d, *J* = 5.7 Hz, 2H), 8.87 (s, 2H). ^13^C NMR (126 MHz, CD_3_OD) δ 26.09 (2C), 26.20 (2C), 26.31 (2C), 28.11 (2C), 30.65 (2C), 34.83 (2C), 50.30 (2C), 51.32 (2C), 55.45 (2C), 115.25 (2C), 117.31 (2C), 127.22 (4C), 128.99 (4C), 130.97 (2C), 138.63 (2C), 143.55 (2C), 147.08 (2C), 149.32 (2C), 157.10 (2C). HRMS-*m/z* (M)^+^ calculated for C_42_H_50_N_6_S_2_: 702.3538, found: 702.3601.

*(1,1’((butane1,4diylbis(4,1phenylene))bis(methylene))bis(4-(azepan-1-yl)thieno[2,3-d]pyrimidin-1-ium)) bromide ****Ff-35***. Following the general procedure, after workup as described previously, compound **Ff-35** was isolated as a brown solid. Yield: 34%, mp: 117–119 °C. ^1^H NMR (500 MHz, CD_3_OD) δ 1.64 (s, 4H), 1.71–1.73 (m, 8H), 1.91–1.93 (m, 4H), 2.06–2.08 (m, 4H), 2.66 (s, 4H), 4.18–4.20 (m, 4H), 4.22–4.24 (m, 4H), 5.64 (s, 4H), 7.26 (d, *J* = 8.2 Hz, 4H), 7.40 (d, *J* = 8.1 Hz, 4H), 7.77 (d, *J* = 6.0 Hz, 2H), 7.85 (d, *J* = 6.0 Hz, 2H), 8.94 (s, 2H). ^13^C NMR (126 MHz, CD_3_OD) δ 27.39 (4C), 28.29 (4C), 31.49 (2C), 35.76 (2C), 51.22 (2C), 52.47 (2C), 58.18 (2C), 117.27 (2C), 124.25 (2C), 124.53 (2C), 129.05 (4C), 129.89 (4C), 130.12 (2C), 144.98 (2C), 148.84 (2C), 155.47 (2C), 157.52 (2C). HRMS-*m/z* (M)^+^ calculated for C_42_H_50_N_6_S_2_: 702.3538, found: 702.3527.

4,4’-(((ethane-1,2-diylbis(oxy))bis(4,1-phenylene))bis(methylene))bis(7-(pyrrolidin-1-yl)thieno[3,2-b]pyridin-4-ium) bromide **Ff-2**. Following the general procedure, after workup as described previously, compound **Ff-2** was isolated as a brown solid. Yield: 21%, mp: 280–282 °C. ^1^H NMR (400 MHz, CD_3_OD) δ 2.19–2.21 (m, 8H), 3.74–3.76 (m, 4H), 4.30–4.32 (m, 8H), 5.63 (s, 4H), 6.74 (d, *J* = 7.4 Hz, 2H), 6.99 (d, *J* = 8.8 Hz, 4H), 7.29 (d, J = 8.8 Hz, 4H), 7.69 (d, *J* = 5.8 Hz, 2H), 8.32 (d, *J* = 5.7 Hz 2H), 8.33 (d, *J* = 7.5 Hz, 2H). ^13^C NMR (101 MHz, CD_3_OD) δ 24.12 (2C), 25.79 (2C), 50.65 (4C), 57.77 (2C), 66.50 (2C), 101.85 (2C), 114.86 (4C), 117.26 (2C), 120.79 (2C), 126.59 (2C), 128.60 (4C), 135.86 (2C), 141.38 (2C), 145.84 (2C), 152.02 (2C), 159.15 (2C). HRMS-*m/z* (M)^+^ calculated for C_38_H_40_N_4_O_2_S_2_: 648.2582, found: 648.2584.

1,1’-(((ethane-1,2-diylbis(oxy))bis(4,1-phenylene))bis(methylene))bis(4-(pyrrolidin-1-yl)thieno[3,2-d]pyrimidin-1-ium) bromide **Ff-8**. Following the general procedure, after workup as described previously, compound **Ff-8** was isolated as a white solid. Yield: 32%, mp: 217–219 °C. ^1^H NMR (500 MHz, CD_3_OD) δ 2.13–2.15 (m, 4H). 2.28–2.30 (m, 4H), 4.01–4.03 (m, 4H), 4.22–4.24 (m, 4H), 4.33 (s, 4H), 5.62 (s, 4H), 7.02 (d, *J* = 8.7 Hz, 4H), 7.38 (d, *J* = 8.6 Hz, 4H), 7.66 (d, *J* = 5.7 Hz, 2H), 8.47 (d, *J* = 5.7 Hz, 2H), 8.86 (s, 2H). ^13^C NMR (126 MHz, CD_3_OD) δ 24.53 (2C), 26.70 (2C), 50.17 (2C), 51.13 (2C), 56.05 (2C), 67.44 (2C), 115.92 (4C), 117.70 (2C), 118.08 (2C), 126.76 (2C), 129.81 (4C), 139.56 (2C), 147.07 (2C), 150.55 (2C), 156.18 (2C), 160.20 (2C). HRMS-*m/z* (M)^+^ calculated for C_36_H_38_N_6_O_2_S_2_: 650.2487, found: 650.2497.

1,1’-(((ethane-1,2-diylbis(oxy))bis(4,1-phenylene))bis(methylene))bis(4-(pyrrolidin-1-yl)thieno[2,3-d]pyrimidin-1-ium) bromide **Ff-4**. Following the general procedure, after workup as described previously, compound **Ff-4** was isolated as a brown solid. Yield: 42%, mp: 193–195 °C. ^1^H NMR (500 MHz, CD_3_OD) δ 2.12–2.14 (m, 4H). 2.26–2.28 (m, 4H), 4.03–4.05 (m, 4H), 4.15–4.17 (m, 4H), 4.36 (s, 4H), 5.60 (s, 4H), 7.05 (d, *J* = 8.8 Hz, 4H), 7.45 (d, *J* = 8.8 Hz, 4H), 7.75 (d, *J* = 5.9 Hz, 2H), 7.93 (d, *J* = 5.9 Hz, 2H), 8.92 (s, 2H). ^13^C NMR (126 MHz, CD_3_OD) δ 24.50 (2C), 26.95 (2C), 50.12 (2C), 50.86 (2C), 58.12 (2C), 67.63 (2C), 116.02 (4C), 118.43 (2C), 124.10 (2C), 124.40 (2C), 125.12 (2C), 130.90 (4C), 149.38 (2C), 154.48 (2C), 155.49 (2C), 160.76 (2C). HRMS-*m/z* (M)^+^ calculated for C_36_H_38_N_6_O_2_S_2_: 650.2487, found: 650.2514.

1,1′-(((ethane-1,2-diylbis(oxy))bis(4,1-phenylene))bis(methylene))bis(4-(piperidin-1-yl)thieno[3,2-d]pyrimidin-1-ium) bromide **Ff-28**. Following the general procedure, after workup as described previously, compound **Ff-28** was isolated as a grey solid. Yield: 25%, mp: 198–200 °C. ^1^H NMR (400 MHz, DMSO) δ 1.76–178-(m, 12H), 4.15–4.17 (m, 8H), 4.28 (s, 4H), 5.66 (s, 4H), 6.99 (d, *J* = 8.6 Hz, 4H), 7.44 (d, *J* = 8.5 Hz, 4H), 7.80 (d, *J* = 5.7 Hz, 2H), 8.65 (d, *J* = 5.7 Hz, 2H), 9.17 (s, 2H). ^13^C NMR (101 MHz, DMSO) δ 23.22 (2C), 25.94 (4C), 49.60 (4C), 54.51 (2C), 66.31 (2C), 114.58 (4C), 114.83 (2C), 117.95 (2C), 126.44 (2C), 129.35 (4C), 139.02 (2C), 147.11 (2C), 149.84 (2C), 155.36 (2C), 158.40 (2C). HRMS-*m/z (*M)^+^ calculated for C_38_H_42_N_6_O_2_S_2_: 678.2800, found: 678.2774.

1,1′-(((ethane-1,2-diylbis(oxy))bis(4,1-phenylene))bis(methylene))bis(4-(piperidin-1-yl)thieno[2,3-d]pyrimidin-1-ium) bromide **Ff-5**. Following the general procedure, after workup as described previously, compound **Ff-5** was isolated as a brown solid. Yield: 41%, mp: >280 °C. ^1^H NMR (500 MHz, CD_3_OD) δ 1.89 (m, 12H), 4.25 (m, 8H), 4.36 (s, 4H), 5.60 (s, 4H), 7.06 (d, *J* = 8.8 Hz, 4H), 7.46 (d, J = 8.8 Hz, 4H), 7.78 (d, *J* = 6.0 Hz, 2H), 7.83 (d, *J* = 6.0 Hz, 2H), 8.89 (s, 2H). ^13^C NMR (126 MHz, CD_3_OD) δ 24.29 (2C), 26.56 (4C), 51.06 (4C), 57.95 (2C), 67.46 (2C), 115.86 (4C), 117.16 (2C), 123.82 (2C), 124.54 (2C), 124.92 (2C), 130.79 (4C), 148.86 (2C), 155.65 (2C), 156.90 (2C), 160.61 (2C). HRMS-*m/z* (M)^+^ calculated for C_38_H_42_N_6_O_2_S_2_: 678.2811, found: 678.2839.

1,1′-(((ethane-1,2-diylbis(oxy))bis(4,1-phenylene))bis(methylene))bis(4-(azepan-1-yl)thieno[3,2-d]pyrimidin-1-ium) bromide **Ff-34**. Following the general procedure, after workup as described previously, compound **Ff-34** was isolated as a brown solid. Yield: 45%, mp: 201–202 °C. ^1^H NMR (500 MHz, CD_3_OD) δ 1.71–1.73 (m, 4H), 1.92–1.94 (m, 4H), 2.08–2.10 (m, 8H), 4.22–4.24 (m, 4H), 4.25–4.27 (m, 4H), 4.33 (s, 4H), 5.64 (s, 4H), 7.03 (d, *J* = 8.8 Hz, 4H), 7.39 (d, *J* = 8.7 Hz, 4H), 7.67 (d, *J* = 5.8 Hz, 2H), 8.48 (d, *J* = 5.8 Hz, 2H), 8.87 (s, 2H). ^13^C NMR (126 MHz, CD_3_OD) δ 27.22 (4C), 29.02 (4C), 51.21 (2C), 52.22 (2C), 56.15 (2C), 67.45 (2C), 115.90 (4C), 116.16 (2C), 118.21 (2C), 126.68 (2C), 129.82 (4C), 139.51 (2C), 147.97 (2C), 150.11 (2C), 158.01 (2C), 160.26 (2C). HRMS-*m/z* (M)^+^ calculated for C_40_H_46_N_6_O_2_S_2_: 706.3124, found: 706.3068.

1,1′-(((ethane-1,2-diylbis(oxy))bis(4,1-phenylene))bis(methylene))bis(4-(azepan-1-yl)thieno[3,2-d]pyrimidin-1-ium) bromide **Ff-36**. Following the general procedure, after workup as described previously, compound **Ff-36** was isolated as a brown solid. Yield: 27%, mp: 205–207 °C. ^1^H NMR (400 MHz, CD_3_OD) δ 1.72–1.73 (m, 8H), 2.00–2.03 (m, 4H), 2.04–2.06 (m, 4H), 4.19–4.22 (m, 4H), 4.22–4.24 (m, 4H), 4.37 (s, 4H), 5.61 (s, 4H), 7.07 (d, *J* = 8.7 Hz, 4H), 7.47 (d, *J* = 8.7 Hz, 4H), 7.78 (d, *J* = 6.0 Hz, 2H), 7.85 (d, *J* = 6.0 Hz, 2H), 8.91 (s, 2H). ^13^C NMR (101 MHz, DMSO) δ 26.31 (4C), 27.00 (4C), 49.82 (2C), 51.23 (2C), 56.51 (2C), 66.35 (2C), 114.84 (4C), 116.08 (2C), 123.83 (2C), 124.35 (2C), 124.45 (2C), 130.11 (4C), 148.46 (2C), 153.91 (2C), 156.10 (2C), 158.73 (2C). HRMS-*m/z* (M)^+^ calculated for C_40_H_46_N_6_O_2_S_2_: 706.3124, found: 706.3183.

### 3.2. Biological Activity

Table 2 summarizes the effect of the final compounds in infected erythrocytes. In general, all compounds show nanomolar activity, the most active compound being **Ff-4** (11.5 nM). In terms of structure, there seems to be no difference between monocationic and bis-cationic compounds regarding their antimalarial activity. Among the monocationic compounds, **Fa-M3** (isomer thieno[2,3-d]pyrimidin) stands out, but the rest of the isomers follow up very closely, and the difference in the activity is not considered to be remarkable.

With respect to the biscationic compounds and considering the 5 types of linker used, it can be observed that when the spacer is bipyridinyl (**Fg-12**, **Fg-17**, **Fg-13**, **Fa-27**, **Fg-32, Fa-26** and **Fg-20**) there is a notable decrease in the activity with respect to their biphenylic counterparts (**Fg-9, Fa-21**, **Fg-14**, **Fa-24**, **Fg-30**, **Fg-10**, **Fa-22**, **Fg-18**, **Fp-1** and **Fp-8**). This difference is more pronounced when the volume of the 4-substituent in the cationic head increases (**Fa-22** and **Fg-18**
*vs* **Fa-26** and **Fg-20**). Conversely, when the substituent is pyrrolidine there is hardly any difference, except for **Fg-13** and **Fg-17**
*vs*
**Fg-14** and **Fa-21,** where there is more than a 10-fold difference in activity.

The remaining biphenyl and bibenzyl, biphenethyl and 1,2-diphenoxyethane linkers feature very similar values. Starting with the biphenyl spacer, the substituent at 4 has a visible influence, with pyrrolidine again giving the best results (see Table 2 **Fa-21**, **Fg-14, Ff-15** and **Ff-4**), the increased volume of this cycloalkylamine from piperidine to azepane gradually decreases the activity. However, it can be considered one of the best families, as there is hardly any difference between the least active **Fg-18** (IC_50_ = 60 nM) and the most active **Ff-21** (IC_50_ = 13.7 nM). It is worth noting that in this family two thieno pyrimidine isomers **Fp-1** and **Fp-8** substituted at 4 by an *N*-methyl-aniline and *p*-Chloro-*N*-methyl-aniline were also synthesized and evaluated in order to compare the results with those where the substituent is a cycloalkylamine. Nonetheless, a drastic decrease in activity was observed. Finally, the thienopyrimidin isomers stand out against the thienopyridin isomers, suggesting that the introduction of a second *N* atom in the cationic head is highly beneficial.

Continuing the SAR analysis, a very similar pattern emerges. Both when the linkers are bibenzyl and biphenethyl, the thieno[2,3-d] pyrimidine isomers stand out against thieno[3,2-d]pyridine in both. See compounds **Fg-15** and **Fg-31** belongs to the bibenzyl family and **Ff-3** and **Ff-35** with biphenethylic spacer. The volume of the pyrrolidine ring is postulated to be the best, even though the differences are not too great.

The 1,2-diphenoxyethane derivatives behave similarly and the introduction of 2 oxygen atoms between the aromatic rings does not seem to affect their activity. In comparison to the reference compounds **10a** and **10k,** which were previously described by our group [8], it can be observed that the homologous compounds **Ff-4** and **Ff-5** feature an improved activity. Both of them belong to the series of bioisosteres derived from thieno[2,3-d]pyrimidin-1-ium, with a pyrrolidine or a piperidine group in position 4, respectively. In the consecutive isomers with azepane **Ff-34** and **Ff-36** (similar **10k** with quinoline), the activity decreases until, rather unexpectedly, **Ff-36** (thieno[2,3-d]pyrimidin-1-ium), which features 6 times less activity than **10k**.

All final compounds were analyzed as *Pf*CK inhibitors (Table 2), and those with a small residual activity had their IC_50_ values calculated. The values obtained show, at first sight, two striking features; the first is that the monocationic compounds, despite their potent antimalarial activity, do not inhibit the choline kinase activity of the enzyme. The second is that the bipyridinic linker derivatives’ (with moderate antimalarial values) are not inhibitors of *Pf*CK.

Starting with the biphenyl linker derivatives, compounds **Fp-1** and **Fp-8** feature the most remarkable IC_50_ values (7.27 ± 0.23 and 4.17 ± 0.48 μM respectively). Interestingly, these two thienopyrimidine isomers have *N*-methyl-aniline and *p*-Chloro-*N*-methyl aniline as substituents at the 4-position of the heterocycle. Therefore these compounds can be considered as bioisosters of compound **RMS932A** which had an IC_50_ of 1.75 μM and where the 7-quinoline ring is replaced by the two thienopyrimidine isomers. The antimalarial activity has also decreased up to 10-fold with respect to **RSM932A:** from 0.0265 μM to 0.1422 μM for **Fp-1** and 0.2816 μM for **Fp-8**. This result points out that the bioisosteric change has not favoured the inhibition or antimalarial activity. However, it appears that there is a direct correlation between both activities suggesting that these compounds may display a similar mechanism of action to the **RMS932A**, i.e., the antimalarial activity is directly related to the inhibition of the enzyme when the substrate is choline, and hence, the *Pf*CK shows activity towards choline kinase and not ethanolamine kinase. The differences in enzyme activity are lower than those in antimalarial activity, and this could be due to the decreased lipophilicity of these new compounds: **Fp-1** (cLogP = 3.413) and **Fp-8** (cLogP = 4.44) while for **RSM-923A** is (cLogP = 7.44). In this same family, however, it is observed that when the substituents in 4 are cycloalkylamines, for example **Fa-22** and **Fg-18**, the IC_50_ of the enzyme activity reaches moderate values of 11 μM, while the antimalarial activity is 0.0514 μM and 0.0604 μM, respectively; hence, an alternative and complementary mechanism could be involved. Only a moderate increase in enzymatic activity seems to be observed when the isomer is thieno[3,2-d]pyrimidine with respect to the other isomer, which has the sulphur closer to the positive charge of the cationic heads.

In comparison with biphenyl family, the introduction of 2 carbon atoms between the aromatic rings (the bibenzyl family), exerts a negative effect on the enzyme inhibition, **Fg-11** and **Fg-16** being the most active compounds of the bibenzyl family with modest IC_50_ values. In addition, compound **Fg-16** was found to be one of the most lethal against the parasite regarding this family, suggesting that the inhibition of the enzyme is at least partially involved for this compound. The addition of 2 more carbon atoms (**Ff-1**, **Ff-7**, **Ff-3**, **Fa-33**, **Ff-6**, **Fa-29** and **Ff-35**) produces a significant increase in enzyme activity, which is also partially reflected in the notable improvement of the antimalarial activity.

Among these two families, the most active compound is **Fa-29**, with an IC_50_ = 4.64 ± 0.20 μM. It is a derivative of thieno[3,2-d]pyrimidin-1-ium with an azepanyl group as substituent and biphenethyl as a linker and shows an antimalarial activity of 55.6 nM, which is not the best value but exhibits a direct correlation with the inhibition values.

It is worth noting that the asymmetrical reference compounds **BR-23** and **BR-25,** with excellent antimalarial values, can be considered bioisosters of the compounds described in this work: **Fg-9**, **Fa-21**, **Fg-14** for **BR-23** and **Ff-1**, **Ff-7** and **Ff-3** for **BR-25**. A 10-fold decrease in antimalarial activity is observed with respect to the reference compounds, but it is nevertheless striking that the synthesized bioisosters do possess enzymatic activity when the substrate is choline, in contrast to **BR-23** and **BR-25**, which only had inhibitory activity when the enzyme-substrate was ethanolamine.

The **BR-31** derivative can also be considered homologous to the compounds **Fp-1** and **Fp-8,** where a 10-fold decrease in parasite lethality has occurred and, also in this case, this could be related to the lack of lipophilicity of the newly synthesized compounds since the enzymatic inhibition values of **Fp-1** and **Fp-8** are quite considerable.

Analyzing the compounds with the linker 1,2-diphenoxyethane homologous to the compounds already reported [8,27], it can be observed that compounds **Ff-34** and **Ff-36** show the lowest residual enzyme activity with values of 52.17% ± 12.43 and 46.39% ± 12.44, which are precisely the homologues of compound **10k** with quinoline and azepane as a substituent in position 4. However, there is a difference with respect to enzyme activity, since for compound **10a**, the mechanism of action must be different as no inhibition of the enzyme in its choline kinase or ethanolamine kinase activity is observed. In contrast, when the head is quinolinium and carries an azepane group in position 4 (**10k**), values are similar to those presented in this work.

Unfortunately, **Fa-M2**, **Fa-M1** and **Fa-M3** showed severe aggregation even at 5 mM concentration. As a result, it was difficult to measure their exact concentration, which is crucial for enzymatic assays.

## 4. Conclusions

The bioisosteric changes introduced in the compounds presented in this work have led to antimalarial compounds whose potency is in the nanomolar range. Albeit not as potent as some of the previously published compounds, these inhibitors show a remarkable uniformity in terms of antimalarial activity. Among all the compounds, those with the thieno[2,3-d]pyrimidine isomer and a pyrrolidine at position 4 of the cationic head **Fg-15** and **Ff-4** stand out, featuring an antimalarial potency of approximately 11 nanomolar. With respect to the spacers, there does not seem to be much influence on the antimalarial activity.

In a first biological approach, the inhibition of the CK enzyme when the substrate used is choline is postulated as one of the mechanisms of action. Further studies could determine if the mechanism of action of these compounds is also related to inhibition of other enzymes involved in the metabolism of PC, such as *Pf*CCT and *Pf*CEPT, which have been validated as antimalarial targets; or to perturbations of choline transport.

In essence, the compounds presented herein can be considered as excellent candidates for further studies since the bioisosteric changes introduced have increased their solubility and therefore their bioavailability while maintaining almost unchanged their antimalarial activity relative to their homologues.

## Data Availability

Not applicable.

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
