# Peer review of "New Compounds with Bioisosteric Replacement of Classic Choline Kinase Inhibitors Show Potent Antiplasmodial Activity"

_pharmaceutics, 2021, doi:10.3390/pharmaceutics13111842_

Round 1

Reviewer 1 Report

The present manuscript entitled “New compounds with bioisisteric replacement of classic choline kinase inhibitors show potent antiplasmodial activity” by Aguilar-Troyano et al involves the synthesis of 41 compounds, their bioactivity assays against P. falciparum and the SAR analyses to determine the influence of the bioisosteric replacement.

The manuscript has an adequate and well-organized introduction, and they developed specified methodologies according to the explained goals of this study.

Furthermore, the manuscript is related to one important life-threatening pathology (Malaria) caused by a parasite that destroys red blood cells and its affect mainly the children under 5 years.

Although, there are some questions about the research design:

  • It will be important to explain why the authors only use these bioisosters among other available chemical structures.
  • Line 118: the authors wrote “In addition, these new derivatives were developed with the aim to determine whether the replacement of quinoline and pyridine as cationic heads of the classical compounds would enable the molecules to adopt the correct geometry for activity”. But according to this statement there is any bioinformatic approach they used to demonstrate this possible structural arrangement. In this case, molecular docking should be helpful to confirm the correct geometry for these compounds (taking into account current active molecules that have the correct geometry).
  • Moreover, the authors said (Line 120) that “Furthermore, the presence of the additional electron pair in the new N and S atoms would lead to an increase in the affinity for the target through the formation of new hydrogen bonds”. It is important to know which will be the possible HB interactions that the authors expect (considering the target). It should be important to identify them to confirm the hypothesis.

The authors explained there is a crystallographic structure available, so it could be possible to use it to predict the results (before the synthesis) or explain the results (to confirm the hypothesis). The protein data bank (PDB) has several complexes that are useful to analyze in order to improve the results. These suggestions would improve the current conclusions.

  • It is not clear which are the control samples that the authors used (current commercial drugs) for the biological assays.
  • Did the authors analyze the presence of Pan-assay interference compounds(PAINS) in their novel chemical compounds?

Finally, the SAR analyses are according to the obtained results and the novel compounds obtained showed an excellent activity in the range of nanomolar.

I encourage the authors to check some typographical mistakes:

Line 26: Plasmonium falciparum should be Plasmodium falciparum

Lines 137, 153, and 158: there should be a space between the unit and the number

I recommend the acceptance of this paper after the corrections will be performed.

Author Response

Reviewer 1

The present manuscript entitled “New compounds with bioisisteric replacement of classic choline kinase inhibitors show potent antiplasmodial activity” by Aguilar-Troyano et al involves the synthesis of 41 compounds, their bioactivity assays against P. falciparum and the SAR analyses to determine the influence of the bioisosteric replacement.

The manuscript has an adequate and well-organized introduction, and they developed specified methodologies according to the explained goals of this study.

Furthermore, the manuscript is related to one important life-threatening pathology (Malaria) caused by a parasite that destroys red blood cells and its affect mainly the children under 5 years.

We wish to thank the reviewer for the kind words and for the positive evaluation of our manuscript

Although, there are some questions about the research design:

  • It will be important to explain why the authors only use these bioisosters among other available chemical structures.

  • Line 118: the authors wrote “In addition, these new derivatives were developed with the aim to determine whether the replacement of quinoline and pyridine as cationic heads of the classical compounds would enable the molecules to adopt the correct geometry for activity”. But according to this statement there is any bioinformatic approach they used to demonstrate this possible structural arrangement. In this case, molecular docking should be helpful to confirm the correct geometry for these compounds (taking into account current active molecules that have the correct geometry).

  • Moreover, the authors said (Line 120) that “Furthermore, the presence of the additional electron pair in the new N and S atoms would lead to an increase in the affinity for the target through the formation of new hydrogen bonds”. It is important to know which will be the possible HB interactions that the authors expect (considering the target). It should be important to identify them to confirm the hypothesis.

The authors explained there is a crystallographic structure available, so it could be possible to use it to predict the results (before the synthesis) or explain the results (to confirm the hypothesis). The protein data bank (PDB) has several complexes that are useful to analyze in order to improve the results. These suggestions would improve the current conclusions.

Thank you for your comments. In the next paragraph on page 3 we detail how thieno pyrimidines have been extensively used as bioisosters with successful results on kinase inhibition. In addition, purines are recently described as antimalarials in reference 3. All this encouraged us to test these compounds.

It is also important to note that these compounds have also been tested on the human enzyme CK and its antiproliferative effects on several tumor cell lines (unpublished data, to be published soon), including both previous and subsequent molecular modeling studies.

Due to the similarity between both human and Plasmodium enzymes, we wanted to make a first approach in this work as antimalarials, including assays also on the PfCK enzyme.

On the other hand, encouraged by the interesting results as antimalarials of the compounds of the present work, we tried in many possible ways to obtain crystals of the complexes between PfCK and our inhibitors, but without any success. We collected many diffraction data sets but in every structure we solved there was no sign of the compound.

As far as structures are concerned, we have done a lot of co-crystallisation tests to make the protein-inhibitor complex, but we have never managed to obtain the structure of the complex. We don't exclude the possibility that we might try again in the future.

Therefore, and in full agreement with the reviewer's comments, it is our intention to conduct further studies that will help us to better understand the mechanism of action of these molecules.

  • It is not clear which are the control samples that the authors used (current commercial drugs) for the biological assays.

Thank you very much for your comments; we have introduced the chloroquine values in Table 2

  • Did the authors analyze the presence of Pan-assay interference compounds(PAINS) in their novel chemical compounds?

Thank you very much for your comments. As mentioned above, we have made a first approach to this work, but we will take these PAINS into account in future studies in the near future.

Finally, the SAR analyses are according to the obtained results and the novel compounds obtained showed an excellent activity in the range of nanomolar.

I encourage the authors to check some typographical mistakes:

Line 26: Plasmonium falciparum should be Plasmodium falciparum

The typo has been corrected

Lines 137, 153, and 158: there should be a space between the unit and the number

Spaces have been added, where needed.

I recommend the acceptance of this paper after the corrections will be performed.

Reviewer 2 Report

In the manuscript entitled “New Compounds with Bioisosteric Replacement of Classic Choline Kinase Inhibitors Show Potent Antiplasmodial Activity”, Aguilar-Troyano et al. report the synthesis of 41 new compounds, bioisosters of known choline kinase inhibitors. The antimalarial activity of the new synthesized compounds was evaluated in infected erythrocytes and correlated with their ability to inhibit Plasmonium falciparum Choline Kinase (PfCK) that, mediating choline uptake, is essential for the development of the parasite.

SARs analysis indicated that the thieno[2,3-d]pyrimidine isomer substituted in 4 by a pyrrolidine is the best scaffold, independently of the linker (i.e. compounds Fg-15 and Ff-4). Nevertheless, while potent compounds with similar good antimalarial activity have been related to the proposed mechanism of action, some of them still show discrepancies.

Malaria is a disease still widely spread in many regions or the world, and resistance development to antimalarials, insecticides and other prevention barriers, request the development of alternative strategies to block parasite progression and infection as that proposed in the present manuscript.

Experiments are appropriate and properly conducted and the conclusions are supported by the data collected by the authors. The text is clear (although some improvements could be made, see below) and the experimental procedures are reported with suitable detail. For all these reasons, I think that the present manuscript is suitable to be published in “Pharmaceuticals” although some points should be addressed before publication.

  • Introduction needs to be amended avoiding repetitions and not clear sentences. Namely, the paragraph at page 2 from line 88 to line 96 “Plasmodium species parasites need to grow and multiply fueled by precursors supplied by the host…” sounds like a repetition of concepts already introduced. It could be removed. At page 3, lines 107-109, the sentence is not clear. In particular when the authors write “the low toxicity described for some of these previous compounds, have led us to evaluate all compounds…” it is not possible to understand to which compounds they refers to. Please rephrase the sentence and/or add references. Moreover, I think that a figure including the relevant, previously synthesized, compounds mentioned in the introduction, should be added. At least, it should contain the structures of reference compounds listed in table 2.
  • Page3, Figure1. The second linker in the box should be drawn as that depicted in the structure at the bottom of the figure. Moreover, the letter near the brackets of the last structure it not readable, please increase its size.
  • Page 4, lines 135-136 “For flash chromatography, Merck silica gel 60 with a particle size of 0.040 – 0.063 mm (230 – 400 135 mesh ASTM) was used: 500 MHz 1H and 126 MHz 13C NMR Varian Direct Drive; a 400 MHz 1H and 101MHz 13C NMR Varian Direct Drive spectrometers at room temperature”. The colon after “was used” should be a full stop. Moreover, “were used” should be added after “room temperature”.
  • In the section “Chemistry”, HRMS are reported as m/z, thus (M)2+ values should be the half of the expected mass.
  • Page 16, Table 2. It would be helpful if Table 2 could contain the chemical structures of the substituents, not just their names. I think it would help in SARs consideration. Moreover, in the table’s caption, it is not clear which data are from Ref. 26 and which from Refs. 7-8,10. Please, specify.
  • Page 15, lines 195-197. In the sentence “The introduction of 2 carbon atoms between the aromatic rings (the bibenzyl family), exerts a negative effect on the enzyme inhibition, being Fg-11 and Fg-16 the most active compounds.” It is not clear how the effect of the linker is considered negative, as compounds Fg-11 and Fg-16 are among the most active. It sound like a nonsence.
  • Page 18, in “Author Contributions” section is reported “crystallizazation studies A. T and E.P”, although any crystal structure is reported/commented in the manuscript.
  • In Refs. 7 and 8 the Title is missing.
  • The manuscript should be revised for typos correction.

Author Response

In the manuscript entitled “New Compounds with Bioisosteric Replacement of Classic Choline Kinase Inhibitors Show Potent Antiplasmodial Activity”, Aguilar-Troyano et al. report the synthesis of 41 new compounds, bioisosters of known choline kinase inhibitors. The antimalarial activity of the new synthesized compounds was evaluated in infected erythrocytes and correlated with their ability to inhibit Plasmonium falciparum Choline Kinase (PfCK) that, mediating choline uptake, is essential for the development of the parasite.

SARs analysis indicated that the thieno[2,3-d]pyrimidine isomer substituted in 4 by a pyrrolidine is the best scaffold, independently of the linker (i.e. compounds Fg-15 and Ff-4). Nevertheless, while potent compounds with similar good antimalarial activity have been related to the proposed mechanism of action, some of them still show discrepancies.

Malaria is a disease still widely spread in many regions or the world, and resistance development to antimalarials, insecticides and other prevention barriers, request the development of alternative strategies to block parasite progression and infection as that proposed in the present manuscript.

Experiments are appropriate and properly conducted and the conclusions are supported by the data collected by the authors. The text is clear (although some improvements could be made, see below) and the experimental procedures are reported with suitable detail. For all these reasons, I think that the present manuscript is suitable to be published in “Pharmaceuticals” although some points should be addressed before publication.

We wish to thank the reviewer for the kind words and for the positive evaluation of our manuscript

  • Introduction needs to be amended avoiding repetitions and not clear sentences. Namely, the paragraph at page 2 from line 88 to line 96 “Plasmodiumspecies parasites need to grow and multiply fueled by precursors supplied by the host…” sounds like a repetition of concepts already introduced. It could be removed. At page 3, lines 107-109, the sentence is not clear. In particular when the authors write “the low toxicity described for some of these previous compounds, have led us to evaluate all compounds…” it is not possible to understand to which compounds they refers to. Please rephrase the sentence and/or add references. Moreover, I think that a figure including the relevant, previously synthesized, compounds mentioned in the introduction, should be added. At least, it should contain the structures of reference compounds listed in table 2.

We agree with the reviewer that some parts of the original text need rephrasing. In the revised version of our manuscript, we have now written the text more carefully in order to avoid repetitions or ambiguities. Moreover, we have included a Figure 2 with the previously synthesized compounds mentioned in the introduction. Moreover we have included the structures of the substituents in Table 1.

  • Page3, Figure1. The second linker in the box should be drawn as that depicted in the structure at the bottom of the figure, the letter near the brackets of the last structure it not readable, please increase its size.

We thank the reviewer for pointing out ways to improve the quality of the figures in the manuscript. We have now amended Figure1 as per the reviewer’s suggestions.

  • Page 4, lines 135-136 “For flash chromatography, Merck silica gel 60 with a particle size of 0.040 – 0.063 mm (230 – 400 135 mesh ASTM) was used: 500 MHz 1H and 126 MHz 13C NMR Varian Direct Drive; a 400 MHz 1H and 101MHz 13C NMR Varian Direct Drive spectrometers at room temperature”. The colon after “was used” should be a full stop. Moreover, “were used” should be added after “room temperature”.

We have made the suggested changes to the text.

  • In the section “Chemistry”, HRMS are reported as m/z, thus (M)2+values should be the half of the expected mass.

We thank the reviewer for the suggestions. In the revised version of the manuscript, we have changed (M)2+  to (M)+ 

  • Page 16, Table 2. It would be helpful if Table 2 could contain the chemical structures of the substituents, not just their names. I think it would help in SARs consideration. Moreover, in the table’s caption, it is not clear which data are from Ref. 26 and which from Refs. 7-8,10. Please, specify.

We thank the reviewer for the suggestions and for pointing out ambiguities in the references. In the revised version of the manuscript, we have added the chemical structures of the substituents, as per the reviewer’s suggestion. Finally, we have more clearly specified the references in the caption of the table.

  • Page 15, lines 195-197. In the sentence “The introduction of 2 carbon atoms between the aromatic rings (the bibenzyl family), exerts a negative effect on the enzyme inhibition, being Fg-11 and Fg-16 the most active compounds.” It is not clear how the effect of the linker is considered negative, as compounds Fg-11 and Fg-16 are among the most active. It sound like a nonsence.

We thank the reviewer for the suggestions; we have changed this sentence in order to claryfy it.

  • Page 18, in “Author Contributions” section is reported “crystallizazation studies A. T and E.P”, although any crystal structure is reported/commented in the manuscript.

We apologize for the confusion. This short part regarding crystallization studies has been removed from the “Author Contributions” section.

  • In Refs. 7 and 8 the Title is missing.

We apologize for the omission. We have now added the title of the two references.

  • The manuscript should be revised for typos correction.

We have revised the entire manuscript for typos correction.